# Predicting Histologic Grade of Meningiomas Using a Combined Model of Radiomic and Clinical Imaging Features from Preoperative MRI

**DOI:** 10.3390/biomedicines11123268

**Published:** 2023-12-10

**Authors:** Jae Hyun Park, Le Thanh Quang, Woong Yoon, Byung Hyun Baek, Ilwoo Park, Seul Kee Kim

**Affiliations:** 1Department of Radiology, Chonnam National University Hospital, Gwangju 61469, Republic of Korea; jenny_94@naver.com (J.H.P.); radyoon@jnu.ac.kr (W.Y.);; 2Department of Artificial Intelligence Convergence, Chonnam National University, Gwangju 61469, Republic of Korea; ltquangdigi@gmail.com; 3Department of Radiology, Chonnam National University Medical School, Gwangju 61469, Republic of Korea; 4Department of Data Science, Chonnam National University, Gwangju 61186, Republic of Korea; 5Department of Radiology, Chonnam National University Hwasun Hospital, Hwasun-gun 58128, Republic of Korea

**Keywords:** meningioma, radiomic features, machine learning, grading, magnetic resonance imaging

## Abstract

Meningiomas are common primary brain tumors, and their accurate preoperative grading is crucial for treatment planning. This study aimed to evaluate the value of radiomics and clinical imaging features in predicting the histologic grade of meningiomas from preoperative MRI. We retrospectively reviewed patients with intracranial meningiomas from two hospitals. Preoperative MRIs were analyzed for tumor and edema volumes, enhancement patterns, margins, and tumor–brain interfaces. Radiomics features were extracted, and machine learning models were employed to predict meningioma grades. A total of 212 patients were included. In the training group (Hospital 1), significant differences were observed between low-grade and high-grade meningiomas in terms of tumor volume (*p* = 0.012), edema volume (*p* = 0.004), enhancement (*p* = 0.001), margin (*p* < 0.001), and tumor–brain interface (*p* < 0.001). Five radiomics features were selected for model development. The prediction model for radiomics features demonstrated an average validation accuracy of 0.74, while the model for clinical imaging features showed an average validation accuracy of 0.69. When applied to external test data (Hospital 2), the radiomics model achieved an area under the receiver operating characteristics curve (AUC) of 0.72 and accuracy of 0.69, while the clinical imaging model achieved an AUC of 0.82 and accuracy of 0.81. An improved performance was obtained from the model constructed by combining radiomics and clinical imaging features. In the combined model, the AUC and accuracy for meningioma grading were 0.86 and 0.73, respectively. In conclusion, this study demonstrates the potential value of radiomics and clinical imaging features in predicting the histologic grade of meningiomas. The combination of both radiomics and clinical imaging features achieved the highest AUC among the models. Therefore, the combined model of radiomics and clinical imaging features may offer a more effective tool for predicting clinical outcomes in meningioma patients.

## 1. Introduction

Meningiomas are the most common primary brain tumors, constituting 13–26% of all intracranial tumors [1]. The World Health Organization (WHO) categorizes meningiomas into three histopathologic grades. Approximately 90% are histologically benign (Grade I), 5–7% are atypical (Grade II), and 1–3% are anaplastic (Grade III). High-grade meningiomas (Grades II and III) are known to have malignant potential and are more likely to recur after complete resection, sometimes requiring adjuvant radiation therapy [2,3]. Also, the survival rate is higher in low-grade meningioma than in high-grade meningioma. Thus, determining preoperative risk factors for higher tumor grades can provide valuable information for both clinicians and patients.

Magnetic resonance imaging (MRI) is the most important imaging technique for the detection and preoperative evaluation of intracranial meningiomas. Previous studies have reported that preoperative MRI is useful for grading meningiomas and evaluating their histopathological characteristics by analyzing imaging findings such as heterogeneous enhancement, marked peritumoral edema, irregular tumor margins, and bone destruction [4,5,6]. Despite these findings, the image patterns of different grades of meningiomas can often mimic each other, and the usefulness of tumor grading using conventional MRI alone remains controversial [7].

Recently, there has been a rising interest in developing quantitative ways to analyze radiological imaging data. Radiomics is one such way that extracts high-throughput data from medical images using pattern-recognizing mathematical and statistical algorithms to determine pixel intensities. In contrast to the conventional clinical imaging features that are assessed visually by radiologists, which are highly subjective and exhibit inter-observer variability, radiomics analysis can provide a quantitative way to interpret many imaging features. These radiomic features have been shown to reflect underlying pathophysiological characteristics. Furthermore, novel radiomic biomarkers can be developed with prognostic or diagnostic value [8]. As both radiomics and clinical imaging features can serve as prognostic biological factors, machine learning models that combine both radiomics and preoperative clinical imaging features may provide an additional benefit for predicting histologic grade.

The aim of this study is to evaluate the feasibility of using radiomics and clinical imaging features in predicting the histologic grade of meningiomas from preoperative MRI.

## 2. Materials and Methods

### 2.1. Patients

We retrospectively reviewed patients who underwent resection for intracranial meningiomas at Chonnam National University Hwasun Hospital (Hospital 1) from April 2016 to September 2021. The inclusion criteria were: (1) histologically confirmed meningioma with a definite grade (according to the 2016 World Health Organization Classification of Tumors of the Central Nervous System) and (2) availability of standard MR scans before any clinical intervention, including biopsy, consisting of T1- and T2-weighted images (T1WI, T2WI), T1-contrast-enhanced (T1-CE) and fluid-attenuated inversion recovery (FLAIR). The exclusion criteria were: (1) ambiguous pathological grade; (2) incomplete MRI sequences and significant motion artifacts on MR scans; (3) irrelevant intracranial disease history; (4) prior history of surgery or treatment before MRI; and (5) an MRI scan that was not performed at our institution. For the external test set, we included patients who underwent preoperative MRI for intracranial meningiomas at Chonnam National University Hospital (Hospital 2) who met the same inclusion criteria in our validation models. Finally, 164 patients from hospital 1 and 48 patients from hospital 2 were included in the study, respectively. The variables were collected from electronic medical records, pathology reports, and radiology reports.

This retrospective study was approved by the Institutional Review Board of Chonnam National University Hospital and was in accordance with the ethical guidelines of the 2008 Declaration of Helsinki. The requirement for written informed consent was waived due to the retrospective nature of the study. 

### 2.2. MRI Protocols

Preoperative MRI studies were performed at two hospitals. At Hospital 1, MR examinations were performed on 3T scanners (Magnetom TimTrio, Skyra, or Vida: Siemens Healthineers, Erlangen, Germany). The imaging protocols were included T1WI (TR/TE = 2400 − 2540 ms/9.4 ms; matrix = 384 × 269), T2WI (TR/TE = 3500 − 3700 ms/100 − 105 ms; matrix = 448 × 311), FLAIR (TR/TE 7000 ms/80 − 96 ms; matrix = 384 × 230), and T1-CE (TR/TE 149 − 164 ms/3 − 4.4 ms; matrix = 480 × 381). A field of view (FOV) of 230 mm × 230 mm, slice thickness of 4 mm, and no gap were applied to all images. Contrast-enhanced MR scans were acquired after administering a bolus injection of 0.2 mL/kg of contrast agent. 

At Hospital 2, MRI examinations were performed using 3T MR scanners (MAGNETOM TimTrio or Vida: Siemens Healthineers, Erlangen, German; Discovery 750; GE Healthcare Chicago, United States; Ingenia CX: Philips, Amsterdam, The Netherlands). The detailed protocols included the following sequences: T1WI (TR/TE = 2000 − 2400 ms/10 − 13 ms; matrix = 320 − 256 × 230 − 287), T2WI (TR/TE = 3000 − 6000 ms/80 − 100 ms; matrix = 400 − 512 × 259 − 400), FLAIR (TR/TE 4800 − 9400 ms/88 − 340 ms; matrix = 256 − 384 × 204 − 264), and T1-CE (TR/TE 287 − 350 ms/2.5 − 4.6 ms; matrix = 320 − 400 × 224 − 321). An FOV of 230 − 240 mm × 230 − 240 mm, a slice thickness of 5 mm, and a gap of 0.5 mm were applied to all images. T1-CE images were acquired after a bolus injection of 0.2 mL/kg of contrast agent. 

### 2.3. Radiologic Evaluation

Two radiologists, with 2 and 20 years of experience, who were blind to the pathological results, reviewed the MR images. Tumor volume, edema volume, and edema-to-tumor volume ratio were measured for all patients included in the study. A semi-automated evaluation of the tumor and peritumoral edema volumes was performed using 3D Slicer software (version 4.11, http://www.slicer.org (accessed on 11 November 2022)). T1-CE images were used to measure the tumor volume, and FLAIR images were used to access the edema volume in all patients. Also, enhancement pattern (homogeneous/heterogeneous), tumor margin (regular/irregular), tumor–brain interface (clear/unclear), and necrosis and dural tail sign (presence or absence) were assessed on MRI. An irregular tumor margin was defined as a tumor that appeared multilobulated or mushroom-shaped. A clear tumor–brain interface was defined as meningiomas with a distinct cerebrospinal fluid (CSF) cleft between the tumor and brain parenchyma. Figure 1a depicts the typical imaging characteristics of a low-grade meningioma, while Figure 1b illustrates those of a high-grade meningioma.

### 2.4. Image Preprocessing 

A schematic showing the process of image processing and machine learning analysis is shown in Figure 2. Image preprocessing was meticulously designed to standardize radiomic feature extraction from various MRI sequences. The initial step in this process was the application of N4 bias correction, a crucial technique employed to correct for low-frequency-intensity non-uniformities, which are common artifacts in MR imaging. This correction was uniformly applied across T1-CE, T1WI, T2WI, and FLAIR images using the 3D Slicer software. The N4 bias correction, adept at addressing magnetic field inhomogeneities, ensured a homogenous intensity distribution across all images. Subsequently, we engaged in the co-registration of images for each patient, utilizing the General Registration (BRAINS) mode within the 3D Slicer’s suite of registration functions. This mode was specifically chosen for its robustness in aligning all MRI sequences to the axial T1-CE sequence, providing a consistent anatomical framework across various imaging modalities. Following co-registration, the next critical step was skull stripping, performed using the SwissSkullStripper extension in 3D Slicer. This tool is particularly effective for accurately removing non-brain tissues from MRI images, thereby significantly enhancing the precision of our analysis by focusing solely on brain tissues and eliminating extraneous confounds. The final phase in our preprocessing workflow was the application of min–max normalization, especially crucial due to the usage of MRI images from three different machines in our study. This normalization process, applied to the region of interest (ROI) in each image, served to standardize pixel intensity values, ensuring a consistent scale for radiomic feature extraction across different scanners. This process, achieved through custom Python scripts, normalized pixel intensity values, ensuring a consistent baseline for feature extraction [9].

### 2.5. Tumor Segmentation and Radiomics Feature Extraction

3D Slicer was used for semi-automated manual segmentation to delineate the region of interest (ROI) associated with meningioma. Tumor boundaries and peritumoral edema were identified from T1WI and FLAIR images, respectively, by employing thresholding and region-growing segmentation algorithms. Radiomics features were extracted utilizing the open-source Python package PyRadiomics, version 3.0.1 [10]. In total, 851 quantitative features were procured, encompassing two categories: original and wavelet-based features. Prior to their incorporation in the classification model, the values of each feature underwent normalization through the application of the min–max scaler.

### 2.6. Feature Selection and Classifier Model Training and Testing

We imported all 851 radiomic features into the PyCaret tool, version 2.3.10 [11]. PyCaret is an open-source, low-code machine learning library in Python that streamlines certain workflows, including feature selection. In this study, we relied on PyCaret for feature selection. The feature selection process was iterated 100 times with the random selection of the training data. Eight features were found to be selected by PyCaret in each of the 100 tests. We then removed each of the 8 features one-by-one and checked if the performance increased, which rendered a final 5 features.

Regarding the clinical imaging features, a univariate analysis was conducted to select significantly correlated features for the machine learning model. Features with *p*-values less than 0.05 were deemed statistically significant in the multivariate analysis and subsequently chosen for inclusion in the model. LightGBM [12] was employed to train the classifier using the top radiomics features, and the categorical Naïve Bayes (CategoricalNB) model [13] was applied for the classification of clinical imaging features. Both models were trained and evaluated on the Hospital 1 data. The data was initially divided into 5 folds with the 4 parts used for training and 1 part for validation. The validation performance was reported as an average of the 5 folds. The radiomics and clinical imaging models that produced the highest performance among the 5 folds were used for the external test set derived from Hospital 2.

To assess the effect of combining the radiomics and clinical imaging models, a fusion model was developed. In the fusion model, the average of the two probabilities from the radiomics and clinical imaging models was taken as the final probability to predict meningioma grade. The performance of the fusion model was evaluated using the Hospital 2 data.

### 2.7. Statistical Analysis

The clinical imaging features were compared between low-grade and high-grade meningioma groups. Categorical variables were described using percentages, while continuous variables were presented as means. Univariate analysis was performed to select significant radiological characteristics within the low-grade and high-grade groups in the training cohort. Student’s *t*-test and the chi-square test were employed for univariate analysis, with a *p*-value less than 0.05 deemed statistically significant. Statistical analysis was conducted using IBM SPSS software, version 28.0 (SPSS Inc., Chicago, IL, USA). In order to assess the performance of predictive models for meningioma grading, several metrics, including area under the receiver operating characteristics curve (AUC), accuracy, sensitivity, and specificity, were calculated for both the test and validation sets. The performance of the fusion mode was compared with the clinical imaging and radiomics models using DeLong’s test. These evaluations were performed by the in-house-built code using the Python programming language (version 3.7.11).

## 3. Results

### 3.1. Patients’ Characteristics

The study included a total of 212 patients, divided into two groups for training and testing, with 164 patients in the training group (Hospital 1) and 48 patients in the testing group (Hospital 2). The training group comprised 66 males and 98 females, with a mean age of 60.4 years (age range 25–85 years), while the testing group consisted of 11 males and 37 females, with a mean age of 54.9 years (age range 22–78 years). In Hospital 1, there were 89 patients with low-grade meningioma and 76 with high-grade meningioma, while in Hospital 2, there were 33 with low-grade meningioma and 15 patients with high-grade meningioma. There were no significant differences observed in WHO grade (*p* = 0.074) between the training and testing datasets (Table 1).

### 3.2. Clinical Imaging Features

In the training group (Hospital 1), significant differences were found between low-grade and high-grade meningioma groups for tumor volume (*p* = 0.012), edema volume (*p* = 0.004), enhancement (*p* = 0.001), margin (*p* < 0.001), and tumor–brain interface (*p* < 0.001) (Table 2).

### 3.3. Radiomics Features

The selectin of radiomics features was executed 100 times, resulting in 100 unique sets of selected features. Eight features were consistently selected across all iterations and, therefore, considered for a further selection. A systematic removal and the evaluation of these eight features yielded a final selection of five radiomics features: original GLSZM small area emphasis, original shape flatness, wavelet-HHL GLSZM gray level non-uniformity, wavelet-HLL first-order mean, and wavelet-LLL first-order interquartile range.

### 3.4. Diagnostic Performance of the Prediction Model

The LightGBM radiomics model showed an average validation accuracy of 0.74 (range: 0.72–0.75) using 5-fold cross validation (CV). Using the external test data from Hospital 2, this model yielded accuracy, AUC, sensitivity, and specificity of 0.69, 0.72, 0.67, and 0.7, respectively. 

The average validation accuracy of clinical imaging features using 5-fold CV was 0.69 (range: 0.63–0.79). Using the external test data from Hospital 2, the clinical imaging model yielded accuracy, AUC, sensitivity, and specificity of 0.81, 0.82, 0.73, and 0.85, respectively. The fusion model that combined the radiomics and clinical imaging models resulted in slightly improved performances, with accuracy, AUC, sensitivity, and specificity of 0.73, 0.86, 0.73, and 0.73, respectively, for the external test data. The results of the radiomics, clinical imaging, and fusion models are shown in Table 2 and Figure 3.

The analysis of DeLong’s test revealed that the AUC of the combined model was significantly higher than that of the radiomics model (*p* = 0.0012) but was similar to that of the clinical imaging model (*p* = 0.31). 

## 4. Discussion

The present study aimed to evaluate the value of radiomics and clinical imaging features in predicting the histologic grade of meningiomas using preoperative MRI. Our results highlight the potential of combining radiomics and clinical imaging features to provide a quantitative way for the preoperative prediction of meningioma grade, which can be crucial for guiding clinical decision-making and patient management.

Generally, meningioma can be easily diagnosed with reasonable confidence using MRI and CT, as they typically appear as well-defined masses with a broad-based dural attachment and show homogeneous enhancement on post-contrast imaging. Beyond this simple diagnosis, researchers have been using non-invasive imaging biomarkers to predict tumor grading, which affects patient treatment decisions and prognosis. In previous studies, several conventional and advanced MRI findings have been identified as suggestive of high-grade meningiomas [4,6,14,15,16,17,18,19,20]. The imaging findings associated with high-grade meningiomas include higher degree of peritumoral edema, intratumoral necrosis, lower ADC values, heterogeneous enhancement, dural tail sign, irregular or poorly defined tumor margins, and blurred or irregular tumor–brain interface. In our study, we identified significant differences between low-grade and high-grade meningiomas in terms of tumor volume, edema volume, enhancement, margin, and tumor–brain interface. These findings are consistent with previous literature. However, there are still limitations in the overall predictive accuracy of these findings. To overcome these limitations and improve diagnostic accuracy, machine learning-based predictive models using MRI radiomic features are being developed.

For the selection of radiomic features, we used the PyCaret tool. PyCaret is a high-level, open-source Python library for machine learning that streamlines the process of creating, comparing, and deploying models. It provides a unified interface for several machine learning libraries, enabling the user to implement a wide range of algorithms. In addition, PyCaret includes a variety of preprocessing techniques, feature engineering, and feature selection methods [11]. The feature selection algorithm in PyCaret is based on three main algorithms: random forest, LightGBM, and correlation. One of the most widely used feature selection methods in machine learning is least absolute shrinkage and selection operator (LASSO). LASSO is a linear regression technique that applies regularization to prevent overfitting and perform feature selection [21,22]. While LASSO is a powerful technique for feature selection in linear regression tasks, PyCaret’s broader range of tools and features may have provided a more versatile and efficient approach to selecting the optimal features and models compared to using LASSO alone. 

Five radiomics features were selected by the PyCaret tool: original GLSZM small area emphasis, original shape flatness, wavelet-HHL GLSZM gray level non-uniformity, wavelet-HLL first-order mean, and wavelet-LLL first-order interquartile range. Original GLSZM small area emphasis is a measure of the distribution of small size zones in the gray level size zone matrix. In the context of meningioma grading, this feature may be relevant as it can help capture differences in tumor cellularity, which can be indicative of the tumor’s aggressiveness. Original shape flatness is a shape descriptor that quantifies the elongation of an object in three-dimensional space. It is calculated as the ratio of the smallest to the largest principal axis of the best-fitting ellipsoid. In meningioma grading, this feature may help distinguish between different tumor shapes that could be related to the tumor’s invasiveness or growth pattern. Wavelet-HHL GLSZM gray level non-uniformity measures the non-uniformity of gray levels in the texture of an image. High values indicate more heterogeneity in the image, which may be associated with varying cell densities or structural variations within the tumor. Wavelet-HLL first-order mean is the average of the pixel intensity values in an image after applying the wavelet transform. The mean value represents the overall intensity of the image, and changes in this feature may reflect differences in tumor contrast, vascularity, or cell density. These variations could be associated with different meningioma grades and help distinguish between them. Wavelet-LLL first-order interquartile range measures the interquartile range of the pixel intensity values in an image after applying the wavelet transform. The interquartile range can provide information about the distribution and variability of the intensity values in the image. In meningioma grading, this feature may capture variations in tumor heterogeneity or tissue properties. In summary, these five radiomics features presumably reflect different aspects of meningioma characteristics, such as shape, texture, and intensity distribution. They are associated with aggressiveness or growth patterns, which may be predictive of meningioma grading and prognosis.

The LightGBM classifier model demonstrated superior performance among radiomics-based predictive models, with an average validation accuracy of the 5-fold CV of 0.74. When applied to external test data from Hospital 2, the radiomics model achieved an accuracy of 0.69, an AUC of 0.72, a sensitivity of 0.67, and a specificity of 0.7. These findings are consistent with several recently published studies that have investigated the use of radiomics data in the grading of meningiomas [2,7,22,23,24,25,26,27]. A systematic review and meta-analysis conducted by Ugga reported that the overall pooled AUC for machine learning models in meningioma grading was 0.88 (95% CI = 0.84–0.93) [28]. 

Although methodological differences among studies, such as variations in patient populations, imaging modalities, feature selection algorithms, classifier models, sample sizes, and external validation, pose significant challenges in making direct comparisons, our study revealed a relatively lower overall predictive value compared to that of the previous research. Several factors might have contributed to these results. First, our training dataset was obtained from multiple MRI machines, whereas most previous studies used data from a single MRI machine. MRIs at Hospital 1, used to obtain radiomics data in this study, were from three different machines. In general, using data from a single MRI machine can yield higher diagnostic accuracy and AUC for machine learning models because the data from a single MRI scanner has less variation than that from multiple MRI scanners. However, the model trained with data from a single MRI scanner may lack generalizability to data from other MRI machines. On the other hand, models trained with radiomics data from multiple MRI machines can lead to a more robust and generalizable machine learning model, as they are trained on a diverse set of imaging data that accounts for variations in imaging parameters, scanner-specific artifacts, and inter-scanner variability. This may result in lower diagnostic accuracy or AUC compared to a model trained on a single MRI machine. The second factor to consider is differences in patient populations. The current study has a higher proportion of high-grade meningioma patients compared to the previously published studies. This is advantageous as it allows us to evaluate the performance of our model in predicting high-grade tumors, which are generally more challenging to diagnose and require more aggressive treatment. In addition, the inclusion of a larger number of high-grade patients provides more robust data for the development of predictive models. The third factor pertains to the disparity in machine learning models. Previous research often employed support vector machine (SVM) or random forest models [22,28]. In contrast, this study utilized LightGBM. This model exhibits exceptional accuracy and speed, making it suitable for handling extensive datasets. Notably, it possesses the ability to handle missing values and outliers, which are frequently encountered in medical datasets. Moreover, its proficiency in managing imbalanced datasets proves advantageous in medical imaging, where certain diseases may have higher prevalence rates. Despite these benefits, LightGBM has several limitations. Overfitting may occur if the model is trained on a small dataset or if too many features are included in the analysis. Additionally, the model’s accuracy can be influenced by hyperparameters such as the learning rate and the number of trees [12,29].

One of the significant findings of our study is the novel integration of radiomics and clinical imaging features to develop a robust and effective model for predicting meningioma grades. Most studies in the literature have attempted to predict meningioma grade using either radiomics or clinical imaging features independently. The combination of radiomics and clinical imaging data resulted in a notable improvement in AUC (0.86) compared to either the radiomics (0.72, AUC) or clinical imaging (0.82, AUC) models. Although the AUC of the combined model was statistically higher than that of the radiomics model but similar to that of the clinical imaging model, this finding underscores the importance of integrating multiple features for enhanced performance in meningioma grading. Our findings are consistent with several recently published studies that have investigated the combination of radiomics and non-imaging clinical data in other areas of meningioma studies. For instance, a study by Joo reported that an imaging-based model that combined interface radiomics and peritumoral edema could predict brain invasion by meningioma and improve diagnostic performance [30]. Similarly, a study by Park demonstrated that integrating radiomics with clinicopathological features significantly contributed to predicting recurrence in patients with grade 2 meningiomas [31]. These studies, along with our findings, suggest that combining radiomics and clinical features has the potential to be a powerful tool, providing additional information beyond what is visible on conventional imaging.

Our study has several limitations. First, we only used retrospective data. Second, the sample size of the testing group was relatively small, which may limit the generalizability of our findings. In addition, our study did not include other MR imaging sequences, such as perfusion-weighted imaging or diffusion-weighted imaging, which might offer complementary information for predicting meningioma grade.

## 5. Conclusions

In conclusion, our study demonstrates the potential of a combined model that incorporates both radiomics and clinical imaging features for predicting the histologic grade of meningiomas using preoperative MRI. By overcoming the limitations of conventional MRI-based grading and reducing subjectivity, our approach can provide valuable information for clinicians and patients in terms of prognosis and management. Future studies could further validate and refine this model using larger, multi-center cohorts and explore the potential of incorporating additional imaging modalities to enhance predictive performance.

## Figures and Tables

**Figure 1 biomedicines-11-03268-f001:**
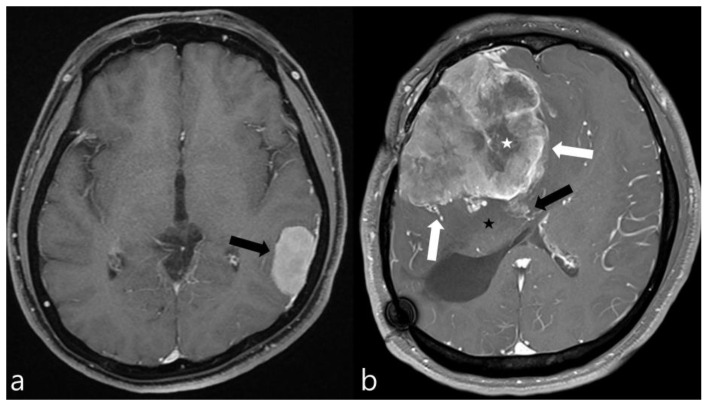
Contrast-enhanced axial MR images of two meningiomas. (**a**) This image exhibits the typical features of a low-grade meningioma, including homogeneous enhancement, smooth margins, absence of peritumoral edema, and no significant mass effect on the surrounding brain parenchyma (black arrow). (**b**) In contrast, this image displays the hallmarks of a high-grade meningioma, characterized by necrotic areas (white asterisk), pronounced mass effect, peritumoral edema (black asterisk), irregular margins (white arrows), and suspicious invasion into the adjacent brain parenchyma (black arrow), indicating a more aggressive tumor behavior.

**Figure 2 biomedicines-11-03268-f002:**
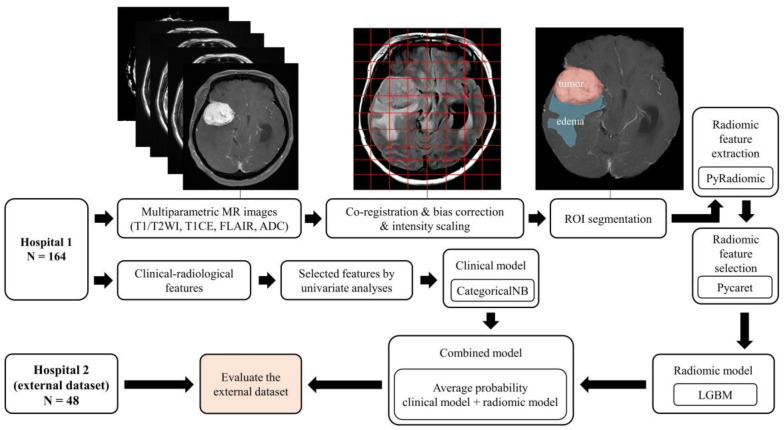
Overview of the image processing and radiomics analysis framework used to develop a machine-learning model.

**Figure 3 biomedicines-11-03268-f003:**
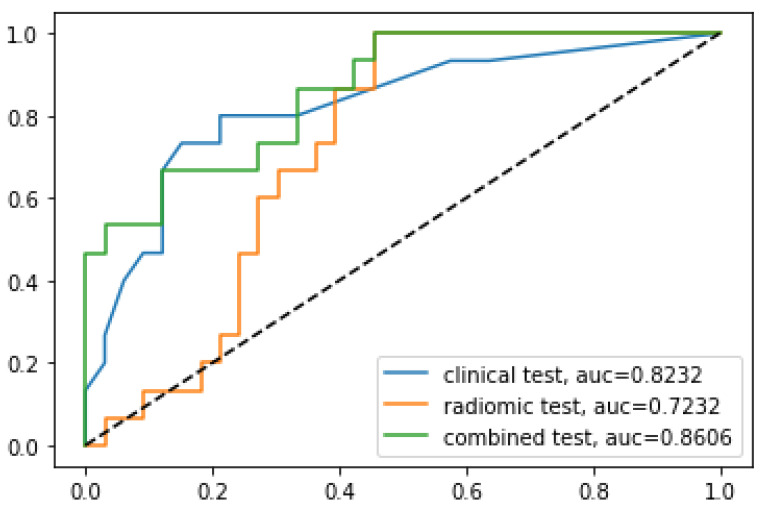
Comparison of receiver operating characteristic curves for the prediction of tumor grading using the external test set from Hospital 2.

**Table 1 biomedicines-11-03268-t001:** Characteristics of patients in the training and testing group.

Variables	Training Set (Hospital 1)			Testing Set (Hospital 2)		
	Low Grade	High Grade	*p*-Value	Low Grade	High Grade	*p*-Value
	(*n* = 89)	(*n* = 75)		(*n* = 33)	(*n* = 15)	
Age	59.9	60.88	0.631	55.14	54.66	0.908
Female sex	56	42	0.368	28	9	0.058
Tumor volume (mm^3^) (mean)	32.5	49.0	0.012	30.35	38.0	0.461
Edema volume (mm^3^) (mean)	24.0	45.1	0.004	41.08	47.25	0.754
Edema/Tumor volume ratio	1.1	1.5	0.433	1.46	1.35	0.859
Enhancement			<0.001			<0.001
Homogeneous	61 (68.5%)	32 (42.7%)		29 (87.9%)	6 (40%)	
Heterogeneous	28 (31.4%)	43 (57.3%)		4 (12.1%)	9 (60%)	
Necrosis			0.046			0.502
Yes	28 (31.4%)	35 (46.7%)		6 (18.2%)	4 (26.7%)	
No	61 (68.5%)	40 (53.3%)		27 (81.8%)	11 (73.3%)	
Dural tail			0.055			0.367
Yes	67 (75.3%)	46 (61.3%)		20 (60.6%)	7 (46.7%)	
No	22 (24.7%)	29 (38.7%)		13 (39.4%)	8 (53.3%)	
Margin			<0.001			0.030
Regular	54 (60.7%)	23 (30.7%)		24 (72.7%)	6 (40%)	
Irregular	35 (39.3%)	52 (69.3%)		9 (27.3%)	9 (60%)	
Tumor-brain interface			<0.001			<0.001
Clear	79 (88.8%)	41 (54.7%)		29 (87.9%)	5 (33.3%)	
Unclear	10 (11.2%)	34 (45.3)		4 (12.1%)	10 (66.7%)	

**Table 2 biomedicines-11-03268-t002:** Performance of machine learning of models for prediction of tumor grading.

Models	Validation Set (Hospital 1)				Testing Set (Hospital 2)			
	AUC	Accuracy	Sensitivity	Specificity	AUC	Accuracy	Sensitivity	Specificity
Clinical imaging	0.77	0.69	0.67	0.89	0.82	0.81	0.73	0.85
Radiomics	0.83	0.74	0.80	0.72	0.72	0.69	0.67	0.70
Combined					0.86	0.73	0.73	0.73

AUC, area under the receiver operating characteristic curve.

## Data Availability

The data supporting the conclusions of this study are available from the corresponding authors on reasonable request.

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
