# Peer review of "Predicting Histologic Grade of Meningiomas Using a Combined Model of Radiomic and Clinical Imaging Features from Preoperative MRI"

_biomedicines, 2023, doi:10.3390/biomedicines11123268_

Round 1
Reviewer 1 Report
Comments and Suggestions for Authors
The authors present an interesting study utilising clinical MRI data obtained from individuals presenting with meningiomas to develop a model aimed towards determining histological grading of same. Briefly, 160+ patient data sets were included and utilised for informing the model, before this applied to another data set obtained from another hospital setting. The individual and combined scoring of different models was found to determine meningiomas scoring to an admirable accuracy, highlighting the potential in developing this model further to assist in the diagnosis of clinical cases.
In reviewing the manuscript, I made a couple of observations. The following should be addressed when preparing a suitable revision.
1. In Figure 1, arrows indicating the areas of interest be of use to the reader
2. The resolution/layout of Figure 2 could be improved to make it more leigible
3. For table 2, the column headings should be expanded upon for clarity
Author Response
Thank you very much for taking the time to review this manuscript. Please find the detailed responses below and the corresponding revisions/corrections highlighted changes in the re-submitted files.
Point-by-point response to Comments and Suggestions for Authors
1. In Figure 1, arrows indicating the areas of interest be of use to the reader
Response: Thank you for your valuable suggestion regarding Figure 1. We have now included arrows and asterisks to highlight the areas of interest. This addition will undoubtedly make the figure more informative and easier to comprehend for our readers.
2. The resolution/layout of Figure 2 could be improved to make it more legible
Response: We appreciate your feedback on Figure 2. As advised, we have enhanced the resolution and layout of this figure to improve its legibility. These modifications will facilitate a better understanding of the depicted data.
3. For table 2, the column headings should be expanded upon for clarity
Response: Thank you for pointing out the need for clearer column headings in Table 2. We have expanded upon these headings to provide more clarity and context, thereby enhancing the table's utility and readability.
Reviewer 2 Report
Comments and Suggestions for Authors
Park et al, submitted the manuscript, "Predicting Histologic Grade of Meningiomas Using a Combined Model of Radiomic and Clinical-Imaging Features from Preoperative MRI" which attempted in the direction of developing a robust modeling that combines radiomics and clinical-imaging features for Meningioma patients.
Minor points that need to be revised by the authors:
1. Please add more description to the image processing procedure, so that other researchers can replicate the same study (if interested).
2. Please focus on how the present study is more impactful in developing quantitative ways to analyze radiological imaging data. Was there any particular model used to compare the statistical results?
3. For the current study, there were patients who have shown different grades of meningioma as the author rightly mentioned in line 201 "In Hospital 1, there were 89 patients with low-grade meningioma and 76 with high-grade, while in Hospital 2, there were 33 with low-grade meningioma and 15 patients with high-grade meningioma". How the other parameters were matched specifically to the one or many patients categorized in one group (either high/low-grade meningioma) and normalized, to conduct such an experiment? Please state clearly, it's a bit confusing in the current form.
Author Response
Thank you very much for taking the time to review this manuscript. Please find the detailed responses below and the corresponding revisions/corrections highlighted changes in the re-submitted files.
Point-by-point response to Comments and Suggestions for Authors
1. Please add more description to the image processing procedure, so that other researchers can replicate the same study (if interested).
Response: We are grateful for your suggestion to elaborate on the image preprocessing procedure. Additional details have been added to this section, ensuring that other researchers can replicate our study more effectively.
2. Please focus on how the present study is more impactful in developing quantitative ways to analyze radiological imaging data. Was there any particular model used to compare the statistical results?
Response: We thank the reviewer for the helpful comment. We have performed the DeLong’s test to compare the AUC of the combined model with that of the clinical-imaging and radiomics models. The analysis of DeLong’s test revealed that the AUC of the combined model was significantly higher than that of the radiomics model (p=0.0012), but was similar to that of the clinical-imaging model (p=0.31). As the reviewer has suggested, we have emphasized the potential of this study in providing quantitative ways to analyze radiological imaging data. The following texts were added/modified to follow the reviewer’s suggestions:
Line 199: “The performance of the fusion mode was compared with the clinical-imaging and radiomics models using the DeLong’s test.”
Line 239: “The analysis of DeLong’s test revealed that the AUC of the combined model was significantly higher than that of the radiomics model (p=0.0012), but was similar to that of the clinical-imaging model (p=0.31).”
Line 270: “Our results highlight the potential of combining radiomics and clinical-imaging features to provide a quantitative way for the preoperative prediction of meningioma grade”
Line 376: “Although the AUC of the combined model was statistically higher than that of the radiomics model, but similar to that of the clinical imaging model, this finding underscores the importance of integrating multiple features for enhanced performance in meningioma grading.”
3. For the current study, there were patients who have shown different grades of meningioma as the author rightly mentioned in line 201 "In Hospital 1, there were 89 patients with low-grade meningioma and 76 with high-grade, while in Hospital 2, there were 33 with low-grade meningioma and 15 patients with high-grade meningioma". How the other parameters were matched specifically to the one or many patients categorized in one group (either high/low-grade meningioma) and normalized, to conduct such an experiment? Please state clearly, it's a bit confusing in the current form.
Response: Thank you for your insightful question regarding the matching and normalization of parameters in our study. We appreciate the opportunity to clarify our methodology. Our study employs an external validation approach, utilizing data from a different hospital as a test set, which is fundamental in machine learning research for assessing model generalizability. Contrary to traditional observational studies, our goal was not to match characteristics between groups but to evaluate the model’s performance on unseen data. This approach allows for a robust assessment of the model’s real-world applicability and generalization ability, vital for deploying AI systems in healthcare settings. Furthermore, the relatively small sample size of our study posed a significant limitation for the implementation of techniques like propensity score matching. Such an approach would have led to a further reduction in our already limited sample, risking inadequate statistical power and potentially compromising the validity of our findings. Our methodology, focusing on data from different hospitals, underscores the model's potential for diverse clinical environments, highlighting its applicability across varied conditions and patient populations. In summary, considering the specific context and objectives of our machine learning study, along with the practical constraints, propensity score matching was neither feasible nor necessary for our analysis.